# LATER SPAN ADAPTATION
# FOR LANGUAGE UNDERSTANDING

## ABSTRACT

Pre-trained contextualized language models (PrLMs) broadly use fine-grained tokens (words or sub-words) as minimal linguistic units in the pre-training phase. Introducing span-level information in pre-training has shown capable of further enhancing PrLMs. However, such methods require enormous resources and lack adaptivity due to huge computational requirements from pre-training. Instead of too early fixing the linguistic unit input as nearly all previous work did, we propose a novel method that combines span-level information into the representations generated by PrLMs during the fine-tuning phase for better flexibility. In this way, the modeling procedure of span-level texts can be more adaptive to different downstream tasks. In detail, we divide the sentence into various span components according to the segmentation generated by a pre-sampled dictionary. Based on the sub-token-level representation provided by PrLMs, we bridge the connection between the tokens in each span and yield an accumulated representation with enhanced span-level information. Experiments on the GLUE benchmark show that our approach remarkably improves the performance of PrLMs in various natural language understanding tasks.

## 1    INTRODUCTION

Pre-trained contextualized language models (PrLMs) such as BERT (Devlin et al., 2018), XLNet (Yang et al., 2019), ELECTRA (Clark et al., 2020) have led to strong performance gains in downstream natural language understanding (NLU) tasks. Such models' impressive power to generate effective contextualized representations is established by using well-designed self-supervised training on a large text corpus. Taking BERT as an example, the model used Masked Language Modeling (MLM) and Nest Sentence Prediction (NSP) as pre-training objects and was trained on a corpus of 3.3 billion words.

PrLMs commonly generate fine-grained representations, i.e., subword-level embeddings, to adapt to broad applications. Different downstream tasks sometimes require representations with different granularity. For example, sentence-level tasks such as natural language inference (Bowman et al., 2015; Nangia et al., 2017), demand an overall sentence-level analysis to predict the relationships between each sentence. There are also token-level tasks, including question answering and named entity recognition, which require models to generate fine-grained output at the token level (Rajpurkar et al., 2016b; Sang & De Meulder, 2003). Therefore, the representations provided by PrLMs are fine-grained (word or sub-word), which can be easily recombined to representations at any granularity, and applied to various downstream tasks without substantial task-specific modifications.

Besides fine-grained tokens and sentences, coarse-grained span-level language units such as phrases, name entities are also essential for NLU tasks. Previous works indicate that the capability to capture span-level information can be enhanced by altering pre-training objectives. SpanBERT (Joshi et al., 2019) extends BERT by masking and predicting text spans rather than a single token for pre-training. ERNIE models (Sun et al., 2019; Zhang et al., 2019a) employ entity level masking as a strategy for pre-training. StructBERT (Wang et al., 2019) encourages PrLMs to incorporate span-level structural information by adding trigram de-shuffling as a new pre-training objective. The methods mentioned above show that the incorporation of span-level information in the pre-training phase is effective for various downstream NLU tasks.

However, since different downstream tasks have different requirements for span-level information, the strategy of incorporating span-level information in pre-training might not be suitable for all downstream tasks. For example, by leveraging entity level masking strategy in pre-training, ERNIE models (Sun et al., 2019; Zhang et al., 2019a) achieve remarkable gain in entity typing and Relation Classification, but when it comes to language inference tasks like MNLI (Nangia et al., 2017), its performance is even worse than BERT. Therefore, incorporating span-level information more flexibly and more universally, is imperatively necessary. The representations generated by PrLMs are supposed to be widely applicable for general cases; meanwhile, they are also expected to be flexibly adapted to various specific downstream tasks. Thus introducing span-level clues in a good timing matters a lot. In this paper, we propose a novel method, **La**ter **S**pan **A**daptapan (LaSA), that would enhance the use of span-level information in a task-specific fine-tuning manner, which is lighter and more adaptive compared to existing methods.

In this work, based on the fine-grained representation generated by BERT, a computationally motivated segmentation is applied to further enhance the utilization of span-level information. Previous work has used semantic role labeling (SRL) (Zhang et al., 2019b) or dependency parsing (Zhou et al., 2019) as auxiliary segmentation tools. Nevertheless, these methods require extra parsing procedure, which reduces the simplicity of use. In our method, the segmentation is obtained according to a pre-sampled $n$-gram dictionary. The fine-grained representation in the same span within the segmentation is aggregated to a span-level representation. On this basis, the span-level representations are further integrated to generate a sentence-level representation to make the most of both fine-grained and span-level information.

We conduct the experiments and analysis on the GLUE benchmark (Wang et al., 2018), which contain various NLU tasks, including natural language inference, semantic similarity, and text classification. Empirical results show that our method can enhance the performance of PrLMs to the same degree as altering the pre-training objectives, but more simply and adaptively. Ablation studies and analysis verify that the introduced method is essential to the further performance improvement.

## 2 RELATED WORK

### 2.1 PRE-TRAINED LANGUAGE MODELS

Learning reliable and broadly applicable word representations has long been a prosperous topic for the NLP community. Language modeling objectives are shown effective for generating satisfying distributed representation (Mnih & Hinton, 2009). By leveraging neural network and large text corpus, Mikolov et al. (2013) and Pennington et al. (2014) achieve to train widely applicable word embeddings in an unsupervised manner. ELMo (Peters et al., 2018) further advances state of the art for various downstream NLU tasks by generating deep contextualized word representations. Equipped with Transformer (Vaswani et al., 2017), GPT (Radford et al., 2018) and BERT (Devlin et al., 2018) further explore transfer learning, where models are firstly pre-trained on a large corpus and then applied to downstream tasks in a fine-tuning manner. Recent PrLMs extends BERT in multiple ways, including using permutation language model (Yang et al., 2019), training on a larger corpus and using more efficient parameters (Liu et al., 2019b), leveraging parameter sharing strategy (Lan et al., 2019), employing GAN-style architecture (Clark et al., 2020). T5 (Raffel et al., 2019) further explores the limit of transfer learning by conducting exhaustive experiments.

### 2.2 COARSE-GRAINED PRE-TRAINING METHODS

Previous works indicate that the incorporation of coarse-grained information in pre-training can enhance the performance of the PrLMs. Initially, BERT uses the prediction of single masked tokens as one of the pre-training objectives. Since BERT uses WordPiece embeddings (Wu et al., 2016), sentences are tokenized into the sub-word level so that the masked token can be a sub-word token such as "##ing". Devlin et al. (2018) then points out that instead of masking a single token, using "whole word masking" strategy can further improve BERT's performance. After that, (Sun et al., 2019; Zhang et al., 2019a) verify that PrLMs can benefit from entity-level masking strategy in pre-training. In SpanBERT (Joshi et al., 2019), the model can better represent and predict spans of text by masking random contiguous spans in pre-training. Recently, by making use of both fine-grained and coarse-grained tokenization, AMBERT (Zhang & Li, 2020) outperforms its precursor in various

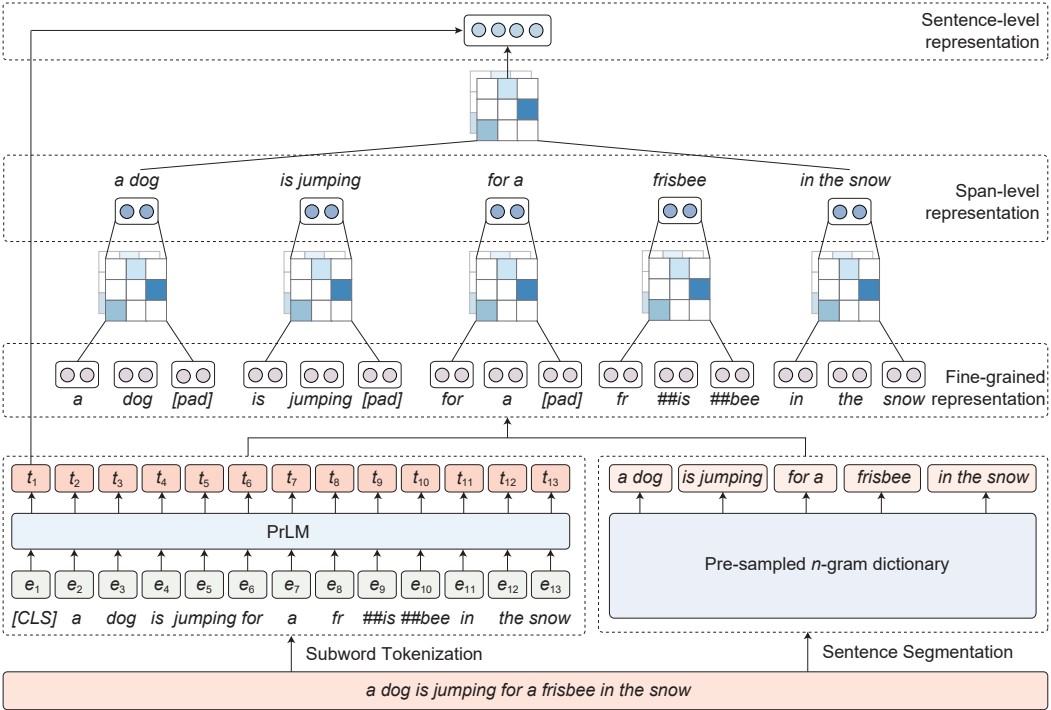

Figure 1: Overview of the framework of our proposed method

NLU tasks. All these works focus on encouraging PrLMs to incorporate coarse-grained information. To the best of our knowledge, incorporating coarse-grained information in fine-tuning is still a white space, which makes our work a valuable attempt.

### 2.3 INTEGRATION OF FINE-GRAINED REPRESENTATION

Different formats of downstream tasks require sentence-level representations, such as natural language inference (Bowman et al., 2015; Nangia et al., 2017), semantic textual similarity (Cer et al., 2017) and sentiment classification (Socher et al., 2013). Besides directly pre-training the representation of coarser granularity (Le & Mikolov, 2014; Logeswaran & Lee, 2018), a lot of methods have been explored to obtain a task-specific sentence-level representation by integrating fine-grained token-level representations(Conneau et al., 2017). Kim (2014) shows that by applying a convolutional neural network (CNN) on top of pre-trained word vectors, we can get a sentence-level representation that is well adapted to classification tasks. Lin et al. (2017) leverage a self-attentive module over hidden states of a BiLSTM to generate sentence-level representations. Zhang et al. (2019b) use a CNN layer to extract word-level representations form sub-word representations and combine them with word-level semantic role representations. Inspired by these methods, after a series of preliminary attempts, we choose a hierarchical CNN architecture to recombine fine-grained representations to coarse-grained ones.

## 3 METHODOLOGY

Figure 1 shows the overview of the framework of our method, which is primarily based on BERT and leverages segmentation as an auxiliary tool. We do not exhaustively illustrate the details of BERT, given the ubiquitousness of the architecture. Further information on BERT is available in Devlin et al. (2018). An input sentence is tokenized to the sub-word level and passed to BERT to obtain the fine-grained representation. In the meantime, the segmentation components of the very input sentence is generated according to a pre-sampled $n$-gram dictionary. Then, we incorporate the segmentation into the fine-grained representation provided by BERT and divide the representation into several spans. After passing the spans through a hierarchical CNN module, we can obtain a

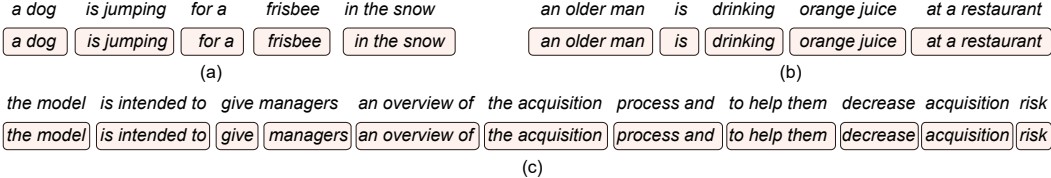

Figure 2: Segmentation Examples

coarse-grained enhanced representation. Eventually, the fine-grained representation of [CLS] token provided by BERT and the coarse-grained information enhanced representation are concatenated to form the final representation that makes the most of multi-grained information for downstream tasks.

### 3.1 SENTENCE SEGMENTATION

Previous works use semantic role labeling (SRL) (Zhang et al., 2019b) and dependency parsing (Zhou et al., 2019) as auxiliary segmentation tools. Nevertheless, these methods require extra parsing procedure, which reduces the simplicity of use. To get a reasonable segmentation in a simpler and more convenient manner, we sample meaningful $n$-grams that occurred in the wikitext-103 dataset based on frequency[1] and obtain a pre-sampled dictionary.

For a given input sentence, we use the pre-sampled dictionary to locate matching $n$-grams from the start of the sentence. Longer $n$-grams are prioritized during the matching procedure. Unmatched tokens would be kept as they are, by doing so, we can obtain a segmentation of the sentence. Figure 2 shows several segmentation examples of the sentences from the GLUE dataset.

### 3.2 SENTENCE ENCODER ARCHITECTURE

For a given input sentence $X = \{x_1, \ldots, x_n\}$ of length $n$, the sentence is first tokenized to sub-word tokens (with a special token [CLS] at the beginning) and transformed to fine-grained representation $E = \{e_1, \ldots, e_m\}$ ($m$ is usually larger than $n$) based on WordPiece embeddings (Wu et al., 2016). Then the transformer encoder (BERT) captures the contextual information for each token via self-attention and produces a sequence of fine-grained contextual embeddings $T = \{t_1, \ldots, t_m\}$, where $t_1$ is the contextual representation of special token [CLS]. According to the segmentation generated by the pre-sampled dictionary, the fine-grained contextual representations are grouped to several spans $\{C_1, \ldots, C_r\}$, with $r$ being a hyperparameter defining the max span number for all processed sentences. Each $C_i$ contains several contextual fine-grained representations extracted from $T$ dedoted as $\{t_1^i, t_2^i, ..., t_l^i\}$, $l$ is another hyperparameter defining the max token number for all the spans. A CNN-Maxpooling module is applied to each $C_i$ to get a span-level representation $c_i$:

$$c_j^i = ReLU(W_1 \left[t_j^i, t_{j+1}^i, \ldots, t_{j+k-1}^i\right] + b_1)$$
$$c_i = MaxPooling(c_1^i, \ldots, c_r^i) \tag{1}$$

where $W_1$ and $b_1$ are trainable parameters and $k$ is the kernel size. Based on the span-level representations $\{c_1, \ldots, c_r\}$, another CNN-Maxpooling module is applied to obtain a sentence-level representation $s$ with enhanced coarse-grained information:

$$s_i' = ReLU(W_2 \left[c_i, c_{i+1}, \ldots, c_{i+k-1}\right] + b_2)$$
$$s = MaxPooling(s_1', \ldots, s_r') \tag{2}$$

Finally we concatenate $s$ with the contextual fine-grained representation $t_1$ of special token [CLS] provided by BERT, and generate a sentence-level representation $s^*$ that make the most of both fine-grained and coarse-grained information for downstream tasks: $s^* = s \diamond t_1$.

---

[1]We have also tried the PMI approach to prune our dictionary, but the result is not competitive.

## 4 EXPERIMENTS

### 4.1 TASKS AND DATASETS

The evaluation of our method is conducted on nine NLU benchmark datasets, including text classification, natural language inference, and semantic similarity. Among them, eight datasets are available from the GLUE benchmark Wang et al. (2018). We also use SNLI Bowman et al. (2015), a widely acknowledged natural language inference dataset, as one of our datasets.

**Classification** Two categories of Classification datasets are involved in our method. The Corpus of Linguistic Acceptability (CoLA) (Warstadt et al., 2018) gives judgment on whether an English sentence is linguistically acceptable or not. The Stanford Sentiment Treebank (SST-2) (Socher et al., 2013) provides sentiment labels, i.e., positive or negative, for sentences extracted from movie reviews.

**Natural Language Inference** Natural Language Inference aims at determining the relationship between a pair of sentences, such as entailment, neutral, and contradiction, based on semantic meanings. Four datasets are involved in our study, including Multi-Genre Natural Language Inference (MNLI) (Nangia et al., 2017), Question Natural Language Inference (QNLI) (Rajpurkar et al., 2016a), Stanford Natural Language Inference (SNLI) (Bowman et al., 2015) and Recognizing Textual Entailment (RTE) (Bentivogli et al., 2009).

**Semantic Similarity** Semantic similarity tasks target at giving a judgment on whether two sentences are semantically equivalent or not. The current challenge for Semantic Similarity is to increase the accuracy of rephrased concepts recognition, negation understanding, and syntactic ambiguity discernment. Three datasets are employed, including Semantic Textual Similarity benchmark (STS-B) (Cer et al., 2017), Quora Question Pairs (QQP) dataset (Chen et al., 2018) and Microsoft Paraphrase corpus (MRPC) (Dolan & Brockett, 2005).

### 4.2 PRE-TRAINED LANGUAGE MODEL

We employ the PyTorch implementation of BERT based on HuggingFace's codebase[2] (Wolf et al., 2019). We use two released BERT versions as our pre-trained encoder as well as our baselines :

**BERT Base** The pre-trained language model was released by Devlin et al. (2018) with 12 layers, 798 hidden layer size, and up to 108 million parameters.

**BERT Large Whole Word Masking** The pre-trained language model released by Devlin et al. (2018) with 24 layers, 1048 hidden layer size, and up to 334 million parameters. The BERT Large wwm employs whole word masking strategy in pre-training and achieves a better performance than the initial version of BERT Large. So we employ BERT Large whole work masking as a stronger baseline, and further show that our method can enhance the utilization of span-level information.

### 4.3 SET UP

We sample all the $n$-grams, with $n \leq 5$, that occur in the wikitext-103 dataset more than ten times to form a dictionary. The dictionary contains more than 400 thousand $n$-grams and is used to segment input sentences. In the process of segmentation, two hyperparameters are introduced: $r$ defining the largest number of spans within an individual sentence, and $l$ representing the largest number of tokens contained within a single span. Padding and tail are applied to guarantee the dimensions of features for each sentence to maintain the same. We set $r$ equals to 16, and based on downstream tasks, choose $l$ in $\{64,128\}$ .

We follow the same fine-tuning procedure as BERT. Sentence-level representation $s^*$ (with enhanced span-level information) is directly passed to a dense layer to get the class logits or score. Training objectives are CrossEntropy for classification tasks and Mean Square Error loss for regression tasks.

---

[2]https://github.com/huggingface/pytorch-pretrained-BERT

| Method | Classification | | Natural Language Inference | | | Semantic Similarity | | | Avg. |
|---|---|---|---|---|---|---|---|---|---|
| | CoLA | SST-2 | MNLI | QNLI | RTE | MRPC | QQP | STS-B | - |
| | (mc) | (acc) | m/mm(acc) | (acc) | (acc) | (F1) | (F1) | (pc) | - |
| | *In literature* | | | | | | | | |
| BiLSTM+ELMo+Attn | 36.0 | 90.4 | 76.4/76.1 | 79.9 | 56.8 | 84.9 | 64.8 | 75.1 | 70.5 |
| GPT | 45.4 | 91.3 | 82.1/81.4 | 88.1 | 56.0 | 82.3 | 70.3 | 82.0 | 72.8 |
| GPT on STILTs | 47.2 | 93.1 | 80.8/80.6 | 87.2 | 69.1 | 87.7 | 70.1 | 85.3 | 76.9 |
| MT-DNN | 61.5 | **95.6** | 86.7/86.0 | - | 75.5 | 90.0 | **72.4** | 88.3 | 82.2 |
| BERT$_{BASE}$ | 52.1 | 93.5 | 84.6/83.4 | - | 66.4 | 88.9 | 71.2 | 87.1 | 78.3 |
| BERT$_{LARGE}$ | 60.5 | 94.9 | 86.7/85.9 | 92.7 | 70.1 | 89.3 | 72.1 | 87.6 | 80.5 |
| | *Our implementation* | | | | | | | | |
| BERT$_{BASE}$ | 51.4 | 92.1 | 84.4/83.5 | 90.3 | 67.1 | 88.3 | 71.3 | 85.1 | 79.3 |
| BERT$_{BASE}$ + LaSA | 55.1 | 93.6 | 84.8/84.3 | 90.6 | 69.6 | 88.7 | 71.9 | 86.5 | 80.4 |
| BERT$_{WWM}$ | 61.1 | 93.6 | 87.1/86.5 | 93.9 | 77.3 | 90.0 | 71.9 | 88.1 | 83.3 |
| BERT$_{WWM}$ + LaSA | **62.9** | 94.1 | **87.6/87.0** | **94.3** | **81.4** | **91.1** | **72.4** | **89.1** | **84.4** |

Table 1: Test sets performance on GLUE benchmark. All the results are obtained from Liu et al. (2019a), Radford et al. (2018). We exclude the problematic WNLI set and do not show the accuracy of the datasets have F1 scores to save space. *mc* and *pc* denote the Matthews correlation and Pearson correlation respectively.

We use Adam as our optimizer with an initial learning rate in {1e-5,2e-5, 3e-5}, a warm-up rate of 0.1, and an L2 weight decay of 0.01. The batch size is set in {16, 32, 48}. The maximum number of epochs is selected in {2,3,4,5} depending on downstream tasks. Sentences are tokenized and transformed to WordPiece embeddings, with maximum length chosen in {128, 256}. The CNN layer's output size is the same as the hidden size of PrLM, and the kernel size is set to 3.

## 4.4 RESULTS

Table 1 shows results on the GLUE benchmark datasets, demonstrating our LaSA approach can remarkably improve the performance of PrLMs on downstream tasks. Since LaSA is primarily established on BERT, and is evaluated under the same procedure, it is evident that the improvement is entirely contributed by our newly introduced span adaptation.

Compared to previous works that demand large-scale pre-training in the incorporation of span-level information, LaSA is able to achieve the same level of performance improvement, and moreover, require remarkably less time and resources. Table 2 shows the comparison of LaSA and SpanBERT on the GLUE benchmark. reaches one percent gain on average over its baseline (BERT-1seq), which is a BERT pre-trained without next sentence prediction (NSP) object. Comparatively, LaSA improves the performance of our baseline by one point one percent on average. Furthermore, LaSA can improve the performance of PrLM on every downstream task, while underperforms its baseline on dataset like MRPC and QQP. Such a comparison indicats that our method can be more dynamically adapted to different downstream tasks.

| Method | Classification | | Natural Language Inference | | | Semantic Similarity | | | Avg. | Gain |
|---|---|---|---|---|---|---|---|---|---|---|
| | CoLA | SST-2 | MNLI | QNLI | RTE | MRPC | QQP | STS-B | - | - |
| | (mc) | (acc) | m/mm(acc) | (acc) | (acc) | (F1) | (F1) | (pc) | - | - |
| BERT-1seq[3] | 63.5 | 94.8 | 88.0/87.4 | 93.0 | 72.1 | 91.2 | 72.1 | 89.0 | 83.5 | 1.0 |
| SpanBERT | 64.3 | 94.8 | 88.1/87.7 | 94.3 | 79.0 | 90.9 | 71.9 | 89.9 | 84.5 | |
| BERT$_{WWM}$ | 61.1 | 93.6 | 87.1/86.5 | 93.9 | 77.3 | 90.0 | 71.9 | 88.1 | 83.3 | 1.1 |
| BERT$_{WWM}$ + LaSA | 62.9 | 94.1 | 87.6/87.0 | 94.3 | 81.4 | 91.1 | 72.4 | 89.1 | 84.4 | |

Table 2: Compairison with on test sets of GLUE benchmark

Table 3 shows our approach also improves PrMLs performance on the SNLI benchmark. The result indicates that the performance of our method is comparable to published SOTA achieved by SemBERT. Unlike SemBERT that incorporates explicit contextual semantics by leveraging a pre-trained semantic role labeler, LaSA only uses a pre-sampled dictionary to provide the auxiliary segmentation information. Thus, compared to SemBERT, LaSA is a more flexible method to achieve performance improvement.

---

[3]The baseline of , a BERT pre-trained without next sentence prediction object.

|       | BERT$_{BASE}$ | BERT$_{BASE}$+LaSA | BERT$_{WWM}$ | BERT$_{WWM}$+LaSA | SemBERT$_{WWM}$ |
|-------|------|------|------|------|------|
| dev   | 90.6 | 91.3 | 92.0 | **92.3** | 92.2 |
| test  | 90.7 | 91.1 | 91.4 | 91.7 | **91.9** |

Table 3: Accuracy on dev and test sets of SNLI. SemBERT$_{WWM}$ (Zhang et al., 2019b) is the published SOTA on SNLI.

## 5 ANALYSIS

### 5.1 ABLATION STUDY

As shown in Table 4, to evaluate the contributions of key factors in LaSA, we perform several ablation studies on the dev sets of eight downstream tasks using BERT$_{BASE}$ as our PrLM. Three sets of ablation studies are conducted. Firstly, in order to see if the hierarchical CNN structure alone can improve the overall performance, we directly apply a hierarchical CNN structure without the segmentation. Secondly, in order to test whether our specific segmentation really helps the model to better incorporate the span-level information, we conduct the experiments with randomly segmented sentences. Thirdly, in order to compare the segmentation generated by our method and the segmentation generated by a pre-trained chunker, we leverage a pre-trained chunker provided by NLTK (Natural Language Toolkit) to generate the spans and conduct the experiments with the same procedure.

According to the result, both hierarchical CNN structure and random segmentation can improve the performance on various downstream tasks to some extent. The result of the third experiments indicate that the segmentation provided by pre-trained chunker is beneficial to our method. However a pre-trained chunker requires extra POS parsing procedure, whereas our method only leverage a pre-sampled dictionary which is more time-saving and achieves better results. Our LaSA method improve the performance on all downstream tasks significantly, raising the average score from 82.6 of baseline to 84.2. This result demonstrates that the improvement is largely due to the specific sentence segmentation of our method.

| Method | Classification | | Natural Language Inference | | | Semantic Similarity | | | Avg. |
|--------|------|------|------|------|------|------|------|------|------|
|        | CoLA | SST-2 | MNLI | QNLI | RTE | MRPC | QQP | STS-B | - |
|        | (mc) | (acc) | m/mm(acc) | (acc) | (acc) | (F1) | (acc) | (pc) | - |
| BERT$_{BASE}$ | 51.7 | 92.0 | 84.0/84.5 | 91.3 | 68.7 | 91.0 | 90.8 | 89.0 | 82.6 |
| BERT$_{BASE}$ + CNN | 53.4 | 91.8 | 84.2/84.4 | 91.2 | 65.0 | **91.6** | **91.2** | 89.7 | 82.5 |
| BERT$_{BASE}$ + Random LaSA[4] | 53.8 | 91.7 | 84.5/84.7 | 91.6 | 70.4 | 90.0 | 91.0 | 89.7 | 83.0 |
| BERT$_{BASE}$ + NLTK LaSA[5] | 58.2 | 92.1 | 84.4/84.8 | **91.8** | 70.0 | 91.3 | 91.0 | 89.8 | 83.7 |
| BERT$_{BASE}$ + LaSA | **59.1** | **93.5** | **84.7/85.1** | 91.7 | **71.1** | 91.2 | **91.2** | **90.0** | **84.2** |

Table 4: Ablation studied on dev sets of GLUE benchmark

### 5.2 ENCODER ARCHITECTURE

Conneau et al. (2017) manifest that different sentence encoder architectures have a significant impact on the performance of models. Toshniwal et al. (2020) also indicate that the choice of span representation has significant impact on many natural language processing (NLP) tasks involve reasoning with textual spans, including question answering, entity recognition, and coreference resolution.

In order to test the efficacy of LaSA's encoder architecture, we respectively substitute the component of the encoding layer and the overall structure. For the component of the encoding layer, we compare CNN (Kim, 2014) with the Self-attentive module (Lin et al., 2017). For the overall structure, two categories are considered: a single layer structure with the max-pooling operation and a hierarchical structure.

---

[4]Random LaSA indicate the LaSA method with randomly segmented sentences.
[5]NLTK LaSA indicate the LaSA method with segmentations generated by NLTK pre-trained chunker.

Mapping each component of the encoding layer with the overall structure, we obtain four different encoder architectures: CNN-Maxpooling, CNN-CNN, Attention-Maxpooling, Attention-Attention. Experiments are conducted on SNLI dev and test sets. As shown in table 5, the hierarchical CNN (CNN-CNN) encoder architecture is most suitable for our situation.

|      | CNN-Max | CNN-CNN | Attention[6]-Max | Attention-Attention |
| ---- | ------- | ------- | ---------------- | ------------------- |
| dev  | 90.9    | **91.3**| 90.7             | 90.8                |
| test | 90.9    | **91.1**| 90.5             | 90.8                |

Table 5: The influences of different encoder architecture

### 5.3 SIZE OF $n$-GRAM DICTIONARY

Given the fact that we use a pre-sampled dictionary to segment the sentences, different sizes of dictionaries will lead to different segmentation. Initially, for sentences without segmentation, every single token is regarded as a span. Figure 3 shows how dictionary size and corresponding segmentation influence the average number of spans in the sentences in the CoLA and MRPC datasets. As the dictionary size enlarges, more $n$-grams are matched and regrouped together, which significantly reduces the average number of spans.

To study the impact of dictionary size on model performance, we conduct experiments on the dev sets of two downstream tasks: CoLA and MRPC. In order to focus on the influence of segmentation, and eliminate the effects of fine-grained representations generated by PrLM, we do not apply the concatenation operation in this experiment. Instead, representations with enhanced span-level information are directly passed to a dense layer to obtain the prediction results. As shown in 4, experiments incorporating the segmentation provided by pre-sampled $n$-gram dictionary achieve better results than the one with random segmentation. Furthermore, the results indicate that middle-sized dictionaries ($20k$ to $500k$) usually achieve better performance. This phenomenon is consistent with intuition, since small-sized dictionaries tend to miss meaningful $n$-grams, while large-sized ones are inclined to group meaningless $n$-grams together unnecessarily.

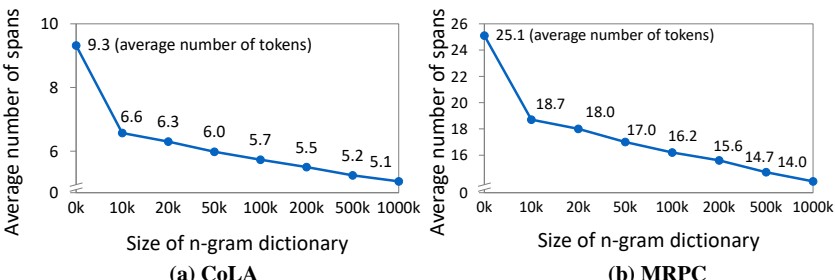

Figure 3: Influence of dictionary size on the average number of spans in the sentences

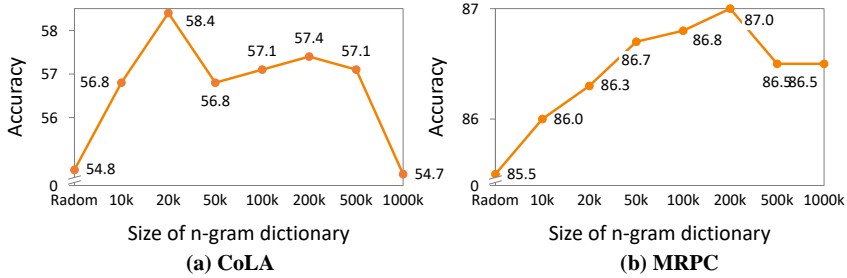

Figure 4: Quantitative study on the influence of the size of $n$-gram dictionary

---

[6]Attention indicate the Self-attentive module (Lin et al., 2017).

## 5.4 APPLICATION TO STRONGER PRLM

Besides BERT, we want to see if our method can improve the performance of stronger PrLMs such as RoBERTa (Liu et al., 2019b), which optimizes BERT with improved pretraining procedure, and SpanBERT (Joshi et al., 2019), which extends BERT by masking and predicting text spans rather than a single token for pre-training.

Table 6 shows that our approach can enhance the performance of both RoBERTa and SpanBERT. For RoBERTa, despite RoBERTa is a very strong baseline, we remarkably improve the performance of RoBERTa on RTE by four percent. For SpanBERT, since it already incorporated span-level information during the pre-training, the results imply that, comparing to SpanBERT, our method utilizes the span-level information and improves the performance of PrLMs in a different manner.

| Method | Classification | | Natural Language Inference | | | Semantic Similarity | | | Avg. |
|---|---|---|---|---|---|---|---|---|---|
| | CoLA | SST-2 | MNLI | QNLI | RTE | MRPC | QQP | STS-B | - |
| | (mc) | (acc) | m/mm(acc) | (acc) | (acc) | (F1) | (acc) | (pc) | - |
| SpanBERT$_{BASE}$ | 55.5 | 92.4 | 85.2/85.1 | 92.1 | 75.1 | 91.2 | 90.9 | 89.4 | 84.0 |
| SpanBERT$_{BASE}$ + LaSA | 57.1 | 92.7 | 85.5/85.4 | 92.6 | 79.4 | 92.1 | 90.9 | 90.1 | 85.1 |
| RoBERTa$_{LARGE}$ | 68.0 | **96.4** | 90.2/**90.2** | **94.7** | 86.6 | 90.9 | **92.2** | **92.4** | 89.0 |
| RoBERTa$_{LARGE}$ + LaSA | **68.9** | 96.1 | **90.3/90.2** | 94.3 | **90.6** | **92.8** | **92.2** | **92.4** | **89.8** |

Table 6: Results on dev sets of GLUE benchmark with stronger baseline

## 5.5 LaSA FOR TOKEN-LEVEL TASK

Our experiments are conducted on the GLUE benchmark which only requires sentence-level representations. However, for natural language understanding, there are other tasks such as named-entity recognition (NER), where token-level representations are needed. Our method can be applied to token-level tasks with simple modification of encoder architecture (e.g. removing the pooling layer of CNN module). Table 7 shows the results of our approach on the CoNLL-2003 Named Entity Recognition (NER) task (Tjong Kim Sang & De Meulder, 2003) with BERT as our PrLM.

The result indicate that our approach can also improve the performance of PrLM on token-level tasks. As discussed by (Toshniwal et al., 2020), the encoder architecture has a remarkable impact on the efficacy of incorporating span-level information. Since we have not explored the encoder architecture for the token-level tasks, we believe that with a more adapted encoder architecture the performances on the token-level tasks can be further promoted.

| | BERT$_{BASE}$ | BERT$_{BASE}$+LaSA | BERT$_{LARGE}$ | BERT$_{LARGE}$+LaSA |
|---|---|---|---|---|
| dev | 91.7 | 92.1 | 92.3 | 92.5 |
| test | 95.7 | 96.2 | 96.5 | 96.8 |

Table 7: F1 on dev and test sets of named entity recognition on the CoNLL-2003 dataset.

## 6 CONCLUSION

This paper proposes a novel method LaSA that takes advantage of flexible span-level information in fine-tuning with fine-grained representations generated by PrLMs. Leveraging a reasonable segmentation provided by a pre-sampled $n$-gram dictionary, LaSA can further enhance the performance of PrLMs on various downstream tasks. Recent works focusing on the incorporation of span-level information in pre-training by altering pre-training objectives or PrLM structures are time-consuming and require enormous resources, therefore, we hope this work could shed light on future studies on adaptation in fine-tuning.

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

## A    APPENDIX

Since fine-tuning BERT on GLUE tasks might have high variance, table 7 report the average results of the application of LaSA to $BERT_{BASE}$ on dev sets of GLUE benchmark. Furthermore, since PrLMs are very strong baselines, the improvements can be marginal, thus a significance test would be beneficial for measuring the difference in model performance.

Follow the procedure of (Zhang et al., 2020), we use the McNemars test to test the statistical significance of our results. This test is designed for paired nominal observations, and it is appropriate for binary classification tasks.

The p-value is defined as the probability, under the null hypothesis, of obtaining a result equal to or more extreme than what was observed. The smaller the p-value, the higher the significance. A commonly used level of reliability of the result is $95\%$, written as p = 0.05. As shown in table 9, comparing to the baseline, for all the binary classification tasks of GLUE benchmark, our method pass the significance test.

| Method | Classification | | Natural Language Inference | | | Semantic Similarity | | | Avg. |
|---|---|---|---|---|---|---|---|---|---|
| | CoLA (mc) | SST-2 (acc) | MNLI m/mm(acc) | QNLI (acc) | RTE (acc) | MRPC (F1) | QQP (acc) | STS-B (pc) | - - |
| $BERT_{BASE}$ | 52.9 | 91.9 | 84.0/84.4 | 91.3 | 68.9 | 91.1 | 90.8 | 88.8 | 82.6 |
| $BERT_{BASE}$ + LaSA | **59.5** | **93.3** | **84.6/85.1** | **91.9** | **71.7** | **91.1** | **91.2** | **90.3** | **84.2** |

Table 8: Results on dev sets of GLUE benchmark, each result is a median over three runs.

| | CoLA | SST-2 | QNLI | RTE | MRPC | QQP |
|---|---|---|---|---|---|---|
| p-value | 0.005 | 0.012 | 0.023 | 0.009 | 0.008 | 0.031 |

Table 9: Results of McNemars tests for binary classification tasks of GLUE benchmark, tests are conducted based on the results of best run.

