# OpenReview forum: "Later Span Adaptation for Language Understanding"
_ICLR.cc/2021/Conference — Reject_

### Official Review · AnonReviewer1 · 2020-10-27
**Is the experimental results significant?**

**Rating:** 6
**Confidence:** 3

**Review:**

Summary:

The paper introduced a fine-tuning approach which adapts the subword level, span level and sentence level to the target tasks. The span segmentation is done by using a pre-trained n-gram statistical model.  Empirical studies on GLUE benchmark show that the proposed approach consistently improves the performance of BERT.

Pros:

1. The idea of leveraging a n-gram model for pre-trained language model fine-tuning is interesting.
2. The proposed method is well motivated and easy to understand.

Cons:

My concern is the significancy of the results. Since the improvement of the proposed method is marginal (~ 1% avg) and fine-tuning BERT on GLUE tasks might have high variance, authors should report the mean and std over multiple runs.

Questions:

If the model is pre-trained with span level information (e.g., spanBERT), will the proposed method outperform normal fine-tuning?

---

> ### Author Response · Authors · 2020-11-16
> **Response to reviewer 1**
>
> Thanks for your interest and constructive comments!
>
> **About the significancy of the results**
> According to your advice, we report the average results on GLUE benchmark dev sets in Appendix. The improvement of 1.6 percent by average is remarkable. Furthermore, we also add the significance test in appendix, results show that the improvements of our method are significant comparing to baseline. In addition, our approach can gain remarkable promotion on certain tasks. We add the experiments with stronger PrLMs, including RoBERTa, in Section 5.4, we can improve the task of RTE by 4.0 percent. This result further proves the efficacy of our method.
>
> **About the application of our method to SpanBERT**
> According to your advice, we add the experiments with SpanBERT, in Section 5.4. The results show that we can further improve the performance of SpanBERT. Such result implies that our method take advantages of the span-level information in a different manner comparing to PrLMs pre-trained with span-level information, which makes our method distinguished comparing to previous works. It is really interesting to find out the origin of this difference, we will do more researches about this topic in the future.

---

### Official Review · AnonReviewer3 · 2020-10-27
**enhancing the performance of PrLMs by span-level information**

**Rating:** 4
**Confidence:** 4

**Review:**

Previous works reveal that span-level information can enhance the performance of PrLMs if they are used in pre-training. However, the existing methods require enormous resources and lack adaptivity. To this end, the paper proposes a method that combines span-level information into the representations generated by PrLMs during the fine-tuning phase. To combine span-level information, the paper first breaks a sentence into various span components. Then, an accumulated representation with enhanced span-level information is built based on the sub-token-level representation provided by PrLMs. The experimental results on the GLUE benchmark show that the proposed method improves the performance of PrLMs. The main contribution of this paper is the introduction of generating the span components via a pre-sampled dictionary. Overall, the proposed method is not novel since similar methods or ideas have been widely used in NPL.
.

Concerns:

1. Employing span-level information in PrLMs was proposed in previous works such as SpanBERT. The paper presented a way that combines span-level information into the representations generated by PrLMs during the fine-tuning phase. However, as shown in Table 2, the proposed method is not better than SpanBERT. So, what are the advantages of the proposed method.
2. As shown in Table 1, compared with the baselines, the proposed method does not bring remarkable promotion.

Minor comments:
1. The subscript “j“ should be “1” in the second line of Formula (1).

---

> ### Author Response · Authors · 2020-11-16
> **Response to reviewer 3**
>
> Thanks a lot for your careful  review. Please see our response below.
>
> **About the main contribution and advantages of this work**
> It is true that previous work such as SpanBERT innovatively pointed out the effectiveness of employing span-level information in PrLMs. Meanwhile, our method remains competitive for the following reasons:
> - **Convenience**:  Previous works incorporate span-level in pre-training, which demands tremendous time and resources. Comparatively, Our method can be directly applied during fine-tuning. The comparison between incorporation of span-level information in pre-training and LaSA is showed below:
> | Methods |   Time  | Resource |
> | :-:| :-: | :-:|
> | pre-training| 32 days | 32 Volta V100 |
> | LaSA | 12 hours max | 2 Titan RTX |
> -  **Task-adaptive**:  For methods that incorporate span-level information in pre-training, the utilization of span-level information is fixed for every downstream task. In our method, the extra module designed to incorporate span-level information is trained during the fine-tuning, resulting in a more dynamically adaptation to different downstream tasks as discussed in Section 4.4.
> -  **Flexible to PrLMs**: We add the experiments with stronger PrLMs in Section 5.4. Our approach can be generally applied to various PrLMs including RoBERTa and SpanBERT.
> -  **Novelty**：As shown by the newly added experiments in Section 5.4, our approach can further improve the performance of PrLMs pre-trained with span level information (e.g. SpanBERT). Such result implies that we our method utilizes the span-level information in a different manner comparing with PrLMs pre-trained with span-level information, which makes our method distinguished comparing with previous works.
>
> **About the improvement**
> Since the PrLMs are very strong baselines, the improvement of one percent is already remarkable. Furthermore, we added the significance test in Appendix, results show that the improvements of our method are significant comparing to the baseline. In addition, our approach can gain a remarkable margin on certain tasks. As you can find in Section 5.5, by applying our method to RoBERTa, which is already a very strong baseline, we can improve the task of RTE by 4.0 percent. This result further proves the efficacy of our method.
>
> **About the typo**
> Thanks a lot for your careful review, the mistake is corrected.

---

### Official Review · AnonReviewer4 · 2020-10-27
**A simple, useful and easy to understand idea. However, the paper lacks novelty and the empirical results may not be convincing.**

**Rating:** 4
**Confidence:** 4

**Review:**

This paper introduces a new mechanism based on span-based representation grouping for better language model finetuning. Specifically, firstly, the paper proposes to utilize an n-gram splitter to segment the text into several segments. Secondly, a hierarchical network is built based on a three-level network (subword level->span level->sentence level). Using these tasks to perform fine-tuning on top of a pre-trained masked language modeling shows improvements than directly fine-tuning on the pre-trained language models. By conducting experiments on the GLUE benchmark with the BERT-base and BERT-large models, the results are improved. The general idea is simple and easy to understand. However, I have several concerns which are list as follows,

1. Combining span-level information has been widely studied in the literature[1] (not cited). Although the tasks between the two papers are different, the novelty of the paper is limited. Besides, it is also worth discussing between Sec 5.2 and [1].
2. The experiments are conducted on BERT-base and BERT-large models. However, the two models are weak baselines. I would like to see the empirical results on stronger models, like RoBERTa related models.
3. As the paper focuses on training span representations, it is necessary to perform experiments on entity-based tasks. The experiments setup may refer to the paper [2].
4. The paper proposed to use the n-grams, it is also necessary to have a comparison with the pre-extracted spans using an off-the-shell chunker.


I also have the following questions:
If a pretrained chunker is utilized to extract chunks, what are the results like?
In section 5.2, does the variant of n-grams affect the results?



For these reasons, I do not recommend acceptance of this paper.




[1] Toshniwal et al. A Cross-Task Analysis of Text Span Representations, ACL RepL4NLP-2020
[2] Yamada et al. LUKE: Deep Contextualized Entity Representations with Entity-aware Self-attention, EMNLP 2020

---

> ### Author Response · Authors · 2020-11-16
> **Response to reviewer 4**
>
> Thanks a lot for your insightful comments. Please see our response below.
>
> **About the literature discussion**
> We have cited this paper and added a discussion in Section 5.2. Since the tasks (and methods) between the two works are very different, as you mentioned the novelty of our work is not affected.
>
> **About stronger baseline**
> Following your advice, we have added the experiments with stronger PrLMs, including RoBERTa, in Section 5.4. Results show that our approach can consistently improve the PrLMs.
>
>
> **About entity-based task**
> Following your advice, we have added the experiments on NER task in Section 5.5. Results show that our approach can also improve the performance of PrLMs on token-level tasks.
>
> **bout pre-trained chunker**
> Following your advice, in Section 5.1, we have added a set of ablation experiments using a pre-trained chunker provided by NLTK to segment the sentences. The results show that a pre-trained chunker can indeed achieve better performances than random segmentation. However, our approach can achieve an even better result than pre-trained chunker, and at the same time be more convenient. Compared with pre-trained chunker, our method processes the data one time faster.

---

### Official Review · AnonReviewer2 · 2020-10-28
**An interesting work on incorporating span information in BERT like models during fine-tuning**

**Rating:** 6
**Confidence:** 4

**Review:**

This paper presents an approach to incorporate span information in pre-trained language models like BERT during fine-tuning. In the proposed approach, the segmentation of a sentence is obtained according to a pre-sampled n-gram dictionary. The fine-grained representation in a same span within the segmentation is aggregated to a span-level representation using a CNN model. These span-level representations are further aggregated using a CNN model to generate sentence-level representation. The experiments show that the proposed model can achieve similar performance gain as other span-based language models which includes span information during pre-training.

The paper is well written and the proposed method is novel. The novelty of the method is in mainly including span information only during fine-tuning and also the way spans are identified in a sentence. The authors conducted an extensive set of experiments with ablation and results show that the proposed idea is effective and attains similar gains as other span-based language model.

It is not clear how this model helps in practice since it is achieving similar performance as other models like SpanBERT. It is true that this model doesn’t introduce complexity while pre-training but pre-training is a one-time process and even if pre-training is slower, it does not affect downstream tasks. However, the proposed model introduces complexity in the downstream tasks and may affect training time in the downstream task.

The authors claim that the proposed model perform similar to models like SpanBERT but it is not clear if both models are improving on similar aspects. Some fine-grained analysis of results with these models might be insightful.

Another concern I have is that the proposed model can only work in the scenarios where sentence-level representation is required. However, there are tasks like named-entity recognition (NER) where word level representation is needed.

---

> ### Author Response · Authors · 2020-11-16
> **Response to Reviewer 2**
>
> Thanks so much for your constructive feedbacks. Please see our response below.
>
> **About the complexity in fine-tuning**
> Yes, our method introduces an extra module for down-stream tasks, the complexity, as well as the computation, is very close to the baseline PrLMs. There are two factors that may influence the complexity and training time: 1) model complexity and 2) sentence segmentation during pre-processing. The total numbers of our model and the baseline are very close – only about 3% extra parameters comparing to the adopted PrLMs. The preprocessing of sentence segmentation is also very fast, about 730 sentences per second per CPU. The processing is parallelizable  and does not affect the whole data processing time. Considering the consistent performance gains over PrLMs on every downstream task, the extra complexity would be acceptable.
>
> Furthermore, for methods that incorporate span-level information in pre-training, the utilization of span-level information is fixed for every downstream task. Whereas, in our method the extra module designed to incorporate span-level information is trained during the fine-tuning, which can be more dynamically adapted to different downstream tasks as discussed in Section 4.4.
>
> **Whether improving the performance on the same aspects comparing with SpanBERT**
> We add the experiments with stronger PrLMs, including SpanBERT, in Section 5.4. The results show that we can further improve the performance of SpanBERT by adopting our later span adaption method. The result implies that our method might take advantages of the span-level information in a different manner compared with PrLMs pre-trained with span level information, such as SpanBERT. It is really interesting to find out the origin of this difference, we will do more researches about this topic in the future.
>
> **About token-level tasks**
> Following your advice, we have added the experiments on NER task in Section 5.5. The results show that our approach can also improve the performance of PrLMs on token-level tasks.

---

### Author Response · Authors · 2020-11-13
**We will update the paper and response soon. Please keep tuned!**

Dear reviewers,

Thanks for all your reviews and waiting. We are working on the revision accordingly and will update the paper with a detailed response soon. Please keep tuned!

---

### Author Response · Authors · 2020-11-22
**Revision Summary**

Dear Reviewers and AC,
We sincerely thank the reviewers for their detailed comments. Reviewers (R2) noted that our method is novel for (a) incorporating span information only during fine-tuning and (b) developing a new way to segment sentences. At the same time, our idea is recognized as well motivated and easy to understand. We have incorporated all the feedback into the newly uploaded version. We briefly summarize the major updates as follows, and please refer to the responses to each reviewer for more detailed explanations:
1.	In Section 5.4, we have added the experiments with stronger PrLMs, including RoBERTa and SpanBERT. The results show that our method can consistently improve the performance of various PrLMs.
2.	In Section 5.1, we have added a set of ablation experiments using a pre-trained chunker provided by NLTK to segment the sentences. The results show that our method can process the data twice as fast as the pre-trained chunker and achieve better performance.
3.	In Section 5.5, we have added the experiments on NER task. Results show that our approach can also improve the performance of PrLMs on token-level tasks.
4.	In Appendix, we report the average results on GLUE benchmark dev sets. We also add the significance test, and the results show that the improvements of our method are significant comparing to the baseline.

Thanks again for your waiting. Please refer to the latest version of the paper for detailed modification. We hope the response and revision can address the concerns. If you have any further comments or suggestions, we will be happy to respond.

---

### Decision · Program_Chairs · 2021-01-07
**Final Decision**

**Decision:**

Reject

**Comment:**

The paper proposes to combine the span-level information into a phrase-level representation in the fine-tuning phrase for pre-trained language models.  The phrases are pre-defined in a dictionary.  Experiments show improvements in various downstream tasks in the GLUE benchmark.  It's a borderline paper.  Various concerns were raised by the reviewers, for example, the relation with the SpanBERT method in pre-training phrase and the significance of the results.  The authors addressed most of the concerns but the reviewers were not fully convinced.  In general, I think it is an interesting paper with good motivation and results.  Hope it can be improved (e.g. more experiments on SpanBERT) and accepted in another conference.